# Resistance to Cassava Whitefly (*Bemisia tabaci*) among Eastern and Southern African Elite Cassava Genotypes

**DOI:** 10.3390/insects15040258

**Published:** 2024-04-09

**Authors:** Wamani Sam, Opio Samuel Morris, Omara Tom, Ocitti Patrick, John Colvin, Omongo Christopher Abu

**Affiliations:** 1National Crops Resources Research Institute, Kampala P.O. Box 7084, Uganda; sammorrisopio@gmail.com (O.S.M.); tom_omara@hotmail.com (O.T.); ocittipatrick@gmail.com (O.P.); caomongo@gmail.com (O.C.A.); 2Natural Resources Institute, University of Greenwich, Chatham Maritime, Kent, Gillingham ME4 4TB, UK; j.colvin@greenwich.ac.uk

**Keywords:** cassava varieties, agro-ecological zones, infestation, sustainable management

## Abstract

**Simple Summary:**

Whitefly is one of the most devastating pests of cassava among smallholder farmers in East and Central Africa. The pest causes severe damage to plants through direct feeding as well as spreading the deadly cassava mosaic disease and cassava brown streak disease. Most of the improved cassava varieties in Uganda were developed for cassava mosaic disease resistance and cassava brown streak disease tolerance. Moreover, few research efforts have focused on ascertaining their reaction to whitefly infestation. A research study was therefore conducted to determine the reaction of several improved regional cassava varieties in a bid to identify those with resistance to whitefly in Uganda. The results reveal that cassava variety Mkumba consistently supports low whitefly populations across all locations. The findings of this study further indicate the potential of improved cassava varieties for the effective and sustainable management of whitefly, especially in the high-pest-pressure areas of Uganda and the continent at large.

**Abstract:**

Cassava whitefly, *Bemisia tabaci*, directly damages cassava leaves by feeding on phloem, causing chlorosis and abscission, leading to a yield loss of up to 50%. The pest also causes indirect damage through sooty mold formation. Most Ugandan cassava varieties resist cassava mosaic disease (CMD) and tolerate cassava brown streak disease (CBSD), but little is known about their response to whitefly infestation. The main objective of this study was to identify cassava genotypes with putative resistance to whitefly in Uganda. This was conducted on 24 improved cassava varieties in three agro-ecological zones during the second rains of 2016. Monthly data were taken for adult and nymph counts, whitefly and sooty mold damage, and CMD and CBSD severities from 2 to 9 months after planting (MAPs). The results show that the whitefly population is highly significantly (*p* < 0.000) amongst varieties across the three agro-ecological zones. Mkumba consistently supported the low adult numbers and nymphs. The findings demonstrate the potential of the improved cassava varieties as sources of whitefly resistance for sustainable management.

## 1. Introduction

Cassava (*manihoti esculenta,* Crantz), a tuberous crop, providing a livelihood for over 500 million people is among the most important food staples worldwide [1]. In Africa, about 70 million people depend on cassava as a major source of carbohydrates [2]. Its tubers can also be processed into various products for use in pharmaceuticals, confectioneries, and breweries [1,3]. In Uganda, cassava is grown by about 29% of agricultural households, producing 4.4 million tons in a land area of 941,000 hectares [4]. Despite the vast economic potential, cassava production in Africa is increasingly under threat by whitefly, *Bemisia tabaci* (Gennadius) (Hemiptera: Aleyrodidae), and millions of smallholder farms have been greatly ravaged by this pest [5]. *Bemisia tabaci* is known to vector cassava mosaic begomoviruses (CMBs) and cassava brown streak geminiviruses (CBSVs), the causative agents of cassava mosaic disease (CMD) and cassava brown streak disease (CBSD) [6,7]. According to [8], the transmission of these cassava viruses causes an estimated yield loss of over USD 1 billion annually in sub-Saharan Africa. In Uganda, during the CMD epidemic in the early 1990s, high whitefly populations were observed in the central and northern parts of the country, which had been devastated by the epidemic [9]. CBSD, which had previously been confined to the East African coastal region, abruptly spread into the mainland of the Great Lakes Region of East and Central Africa in 2004 [8,10]. As reported by [7], the major factor responsible for the spread of this viral disease was the unprecedented increase in the abundance of the whitefly species *Bemisia tabaci* (Gennadius) (Homoptera: Aleyrodidae). To combat these diseases, CMD-resistant and CBSD-tolerant varieties were developed and deployed by the national cassava breeding program in Uganda. However, these varieties have increasingly become susceptible to whitefly infestation and damage.

Whitefly also causes direct damage to cassava by feeding on the phloem of leaves, inducing leaf chlorosis and abscission, which can result in a considerable yield loss of up to about 50% in susceptible varieties [11]. In addition, the honeydew excreted by *Bemisia tabaci* supports sooty mold formation, which decreases the photosynthetic capacity of the cassava plant [12]. According to previous studies conducted among the Latin American and Ugandan landraces in Uganda, it was discovered that Ecu 72, Ofumba Chai, Nabwire 1, and Njule red genotypes exhibited good tolerance levels to cassava whitefly [13]. However, some of these varieties have become susceptible to cassava mosaic disease (CMD) and whitefly infestation over the years. This situation is worsened by the limited research efforts focusing on controlling the pest directly. Moreover, the injudicious use of chemical pesticides for whitefly control increases the production costs, has adverse effects on the ecosystem, and is uneconomical for small-scale farmers [14]. Therefore, stable host plant resistance offers a practical low-cost and long-term solution for sustainable whitefly management in Uganda. This, therefore, called for cassava genotypes with combined disease and whitefly resistance. As such, it is imperative that the research taps into the wider genetic cassava germplasm, of which the best research lines would be into the cassava breeding scheme for the introgression of genes of resistance to whitefly in Uganda. Therefore, in 2016, a study was conducted to evaluate the reaction of 24 improved cassava varieties, which were sourced from five different countries in the region (Uganda, Kenya, Tanzania, Malawi, and Mozambique) in a bid to identify cassava genotypes with putative resistance to whitefly in Uganda. 

## 2. Materials and Methods

### 2.1. Source of Planting Materials

These materials were introduced in Uganda from Kenya, Tanzania, Malawi, and Mozambique as tissue culture plantlets. The materials were hardened at the National Crops Resources Research Institute (NaCRRI) and then established at the cassava breeding multiplication center at RwebiZARDI in Kabarole district, western Uganda, for multiplication. This site is on a high altitude and predominantly a tea-growing area with limited cassava cultivation, and thus has a low disease and whitefly pressure. Materials from this site are routinely monitored and no incidence of both viral diseases has been recorded. 

### 2.2. Experimental Sites

The trial was established in the second rains of 2016 in the Northwestern Savannah Grassland (Lira—N 0229.812, E 03291.879), Kyoga Plains (Kamuli—N 0081.056, E 03312.402), and Lake Victoria Crescent (Wakiso—N 0051.917, E 03263.679) agroecological zones. These sites were selected for the study based on their differential whitefly population pressure and their known history of cassava production in Uganda.

Lake Victoria Crescent is characterized by sandy clay alluvial soils with moist semi-deciduous forest, savannah, and swamps. The area receives bimodal rains in the range of 1750–2000 mm from April to May and October to December for the first and second rains, respectively. The average temperature is between 11 °C and 33 °C. The climate is warm and wet with a high relative humidity and an altitude 1134 m above sea level. Northwestern Savannah Grassland is, however, made up of ferruginous sandy loam soils with intermediate savannah grassland and scattered trees. It also has a bimodal rainfall pattern in the range of 1340–1371 mm. This is followed by a dry spell for about 5 months with a temperature and altitude of 15–25 °C and 951–1341 m above sea level, respectively.

Similar to the Lake Victoria Crescent, the Kyoga Plains agroecological zone is characterized by sandy clay alluvial soils with moist semi-deciduous forest, savannahs, and swamps. The area, however, experiences bimodal rains at 1215–1328 mm from March to May (first rains) and October to December (second rains). The temperature is in the range of 15–32.5 °C. It is generally warm and wet with relatively high humidity, with an altitude 1134 m above sea level.

### 2.3. Experimental Layout and Design

The trial was laid out in the RCBD with a plot size of 5 m by 3 m in 3 replications using 24 cassava genotypes (Table 1). Stem cuttings 20 cm long with about 4 nodes each were planted at a spacing of 1 m × 1 m. In each replication, plots were separated by 2 m from each other and 3 m between replications. The plants were weeded manually using hand hoes to minimize the impact of weeds and alternate hosts of the pests on them. No plant protection measures, including pesticide application, were applied.

### 2.4. Data Collection

Monthly data were recorded 2–9 months after planting (MAPs) on the following parameters: adult and nymph counts, whitefly feeding and sooty mold damage, CMD, and CBSD severity.

#### 2.4.1. Whitefly Adult and Nymph Abundance

Adult whitefly populations were assessed on the top 5 fully expanded apical leaves of 10 randomly selected plants in a plot. Each leaf was gently turned by the petiole to expose the adults on the underside [15]. These were counted manually using a tally counter.

Nymphs were counted on the 14th leaf [16]. This was with the exception of 2 MAPs where the lower mature leaves with the highest nymph numbers were assessed. The assessment was performed on the same 10 plants sampled for the adult whiteflies with the aid of a ×10 magnifying lens.

#### 2.4.2. Whitefly Feeding Damage

Ten plants per plot were visually inspected using a severity scale of 1–5, where 1: no leaf damage, 2: >25% of leaves damaged, with mild chlorosis on few apical leaves, 3: >25–<50% leaves damaged with mild chlorosis and are curled and twisted, 4: >50–<75% of leaves damaged with moderate chlorosis and or wilting, and 5: >75% leaves damaged with defoliation [17]. 

#### 2.4.3. Sooty Mold Severity

This was assessed on 10 plants using a severity scale of 1–5, where 1: no leaf soot on leaves, 2: <25% of plant covered with soot, 3: >25–<50% of plant covered with soot, 4: >50–<75% of plant covered with soot, and 5: >75% of plant covered with soot [17].

#### 2.4.4. Cassava Mosaic and Cassava Brown Streak Disease Severity

Cassava mosaic disease severity was scored on a scale of 1–5, where 1: unaffected shoots or no symptoms observed, 2: mild chlorotic pattern on most leaves and mild distortions at the bases of most leaves, while the remaining parts of the leaves and leaflets appear green and healthy, 3: pronounced mosaic pattern on most leaves and narrowing and distortion of the lower one-third of leaves, 4: severe mosaic distortion of two-thirds of most leaves, general reduction in leaf size, and some stunting of the shoots, and 5: very severe mosaic symptoms on all leaves, distortion, twisting, and severe leaf reduction in most leaves accompanied by severe stunting of plants [18]. The assessment was performed on all 24 plants per plot. 

Cassava brown streak disease foliar severity was assessed in the entire plot using a severity scale of 1–5, where 1: no apparent symptoms, 2: foliar mosaic/chlorosis, but no stem lesions, 3: foliar mosaic, mild or moderate stem lesions, and no dieback, 4: foliar mosaic, severe stem lesions or wilting, but no dieback, and 5: defoliation, severe stem lesions, and dieback [19].

Both CMD and CBSD incidences were calculated as a percentage of the infected plants over the total number of sampled plants per plot. 

### 2.5. Data Analysis

The data were analyzed using the R Version 3.5.1 statistical package. Whitefly adults and nymph counts from 2 to 5 MAPs were subjected to the GLM (generalized linear model) using the family quasi-Poisson since the data were over dispersed. The data were subjected to ANOVA followed by mean separations using the Tukey student test (*p* ≤ 0.05). The means followed by the same letter are not significantly different (*p* ≤ 0.05). Means and standard errors were calculated for damage and sooty mold, CMD, and CBSD incidence.

## 3. Results

### 3.1. Adult Whitefly Population on Different Cassava Genotypes 

There was a significant variation in the population of the adult whiteflies between the three locations (F = 325.8, df = 369.3, *p* < 0.0001) and genotypes (F = 1.859, df = 288.8, *p* < 0.000). Mean adult whitefly populations were generally low in Lira compared to Wakiso and Kamuli. In Kamuli, genotypes Mkumba and KBH/2006/026 harbored the lowest adult whitefly numbers. The lowest populations were also observed on Mkumba and CH05/203 in Wakiso while, in Lira, NAROCASS1 and Mkumba were the least-preferred genotypes by the adult whiteflies. Generally, Mkumba consistently registered the lowest mean adult whitefly population across the three locations (Figure 1).

### 3.2. Nymph Population on Different Cassava Genotypes

The mean nymph population varied significantly between locations (*p* < 0.0001) and genotypes (*p* = 0.001). The least nymph population was recorded on genotypes Mkumba and KBH/2006/26 in Kamuli while, in Wakiso, CH05/203 and Mkumba registered very low populations. Lira, on the other hand, had genotypes NAROCASS 1, Mkumba, and NASE 3 with the least nymphal preference. Mkumba consistently maintained very low nymph numbers across the three locations. Of these locations, Lira recorded the lowest nymph populations (Figure 2).

### 3.3. Whitefly Damage and Sooty Mold Severity on the Different Cassava Genotypes

Across the three locations, average whitefly damage severity among the cassava genotypes ranged from a score of 1 in Lira to 2 in Wakiso. No whitefly or sooty mold damage were observed on any genotype in Lira (Table 2). The lowest mean whitefly damage in Wakiso was observed on genotypes Kalawe, Kizimbani, Sauti, Shibe, and Tajirika. In Kamuli, however, no damage symptoms were scored on CH05/203, Eyope, F19, F10-30-R2, Kalawe, KBH/2006/26, Kizimbani, LMI/2008, Mkumba, NAROCASS 1, NASE 3, NZIVA, Okuhumelela, Olera, Sagonja, Sauti, and Yisazo (Table 2).

Sooty mold damage was not evident on CH05/203 and NASE 14 in Wakiso. On the contrary, very-low sooty mold damage was recorded on Kalawe, F10-30-R2, KBH/2006/26, Olera, Kizimbani, and Mkumba in Kamuli. Sooty mold severity was generally higher in Kamuli than in Wakiso (Table 2). 

### 3.4. Cassava Mosaic Disease Incidence

Cassava mosaic disease symptoms were not observed on Mkumba, NASE 14, NASE 18, or KBH/2006/26 in Wakiso district. Except for Colicanana, Eyope, KBH/2002/066, Kizimbani, Nziva, and Sagonja, no disease symptoms were observed on any other genotype in Lira. Cassava genotypes CH05/203, KBH/2006/26, LMI/2008, NAROCASS 1, NASE 14, NASE 18, Tajirika, and Yisazo did not exhibit any disease symptoms in Kamuli. Largely, the lowest CMD incidence was observed in Lira and the highest in Wakiso (Table 3).

### 3.5. Cassava Brown Streak Disease Incidence

Foliar cassava brown streak disease symptoms were not observed on genotypes Mkumba and Okhumumelela in all three locations. Generally, in Kamuli and Lira, most of the cassava genotypes did not exhibit any foliar symptoms of the disease. This was with the exception of CH05/203, Eyope, F19, Nziva, Olera, and Tajirika in Kamuli, and Colicanana, Kalawe, Sauti, and CH05/203 in Lira. The disease was least prevalent among genotypes in Lira as compared to Wakiso (Table 3).

## 4. Discussion

The study revealed a variation in the whitefly population in the three locations. This variation could be attributed to the different weather factors in these areas. High temperature and relative humidity are key factors that influence whitefly populations. These factors increase the *B. tabaci* rate of development by reducing the time the nymphs take to complete the full life cycle. Kamuli and Wakiso, which had high whitefly numbers, are surrounded by water bodies and thus experience high relative humidity and temperature compared to Lira, with a high temperature but low relative humidity. A similar study conducted by [20] revealed that temperature and relative humidity were crucial for the development of the whitefly (*B. tabaci*) population on tomatoes. They further performed a multiple regression analysis that showed that temperature, relative humidity, rainfall, and sunshine hours combined were responsible for 89% of the variation in the whitefly population observed. In addition, a study carried out by [21] showed that the high adult *B. tabaci* populations were linked to high temperatures and low annual average rainfall around the cassava fields on the Tanzania coast. Land use pattern is another factor that could have been responsible for the difference in the whitefly population structure observed in the three locations. Kamuli and Wakiso districts have a lot of natural resources, like forests, swamps reclaimed for human settlement, and economic activity, and thus the habitats for the natural enemies of cassava whitely could have been destroyed and their population greatly reduced, almost to extinction. The reverse is true for the Lira district where most of the forest cover is still intact, and thus harbors the predators and parasitoids that are proven to effectively control the whitefly population. This could explain the very low adult whitefly and nymph numbers observed in Lira as compared to Kamuli and Wakiso. The observation is supported by the study carried out by [22] in the southern part of France, which showed that undisturbed vegetation cover harbored more predatory mirid bugs that are natural enemies of several pests of tomato. A study carried out by [23] further demonstrated that vegetation cover was among the factors responsible for colonization by *Dicyphus tamaninii* and *Macrolophus caliginosus*, which are known predators of the greenhouse whitefly (*Trialeurodes vaporariorum)* on tomatoes in Spain.

Cassava genotypes reacted differently to the adult whitefly and nymph population infestation. This unearths the host plant resistance mechanisms possessed by these cassava genotypes. Host plant resistance can be categorized into antixenosis, antibiosis, and tolerance [24]. Antixenosis (non-preference) involves the use of inherent morphological traits, like hairiness, plant architecture, and leaf thickness, to help repel heavy pest infestations. Antibiosis, on the other hand, employs the bio-chemical attributes that alter the development and survival of the pest. Lastly, tolerance is where the plant can grow and substantially yield well, even amidst heavy pest colonization. 

Plant morphological traits, like the presence of glandular trichomes (hairs), have been reported as one of the main factors responsible for resistance toward small sucking pests conferred by hindering their egg-laying ability and feeding mechanisms in watermelon cultivars [25]. These glandular trichomes, especially type IV, have been reported to influence whitefly (*B. tabaci*) colonization, depending on their angle on the leaf surface, type, and length on tomato plants [26]. This finding is in agreement with [27], who suggested that plant morphology and biochemistry play a key role in determining the suitability for the growth and development of *B. tabaci* populations. According to [28], cassava defense compounds, like flavonoids, cyanogenic glucosides, and hydroxycoumarins, impact the population of phloem feeders, like *B. tabaci.*

Research carried out by [17] indicated that antibiosis-linked resistance to whitefly in cassava was mainly influenced by biochemical and anti-nutrient compounds, like free sugars, phenolics, and free proteins. These phenolic compounds have been reported to repel the feeding ability of *B. tabaci* on the plant as well as impact the development, behavior, and growth of insects [29,30].

Adult whitefly and nymph populations, cassava age, and leaf morphology are the main drivers of whitefly-associated feeding damage in cassava. In this study, Kamuli and Wakiso, which harbored high whitefly numbers, exhibited the highest levels of damage compared to Lira, where no signs of damage were observed at all. Studies carried out by [31] on sweet potato whitefly on squash revealed that whitefly nymph and adult density values were positively correlated to damage. Across the three locations, whitefly damage peaked at 3 months after planting. At this time, the plant has young tender leaves that are preferred by the vast whitefly populations that have already colonized the plant. Research by [32] also confirmed a similar observation on lemons.

In addition, leaf morphology, especially hairiness, could have influenced the levels of feeding damage observed on the different genotypes. Earlier findings by [33] on cotton revealed that cultivars with a smooth leaf surface supported a high whitefly population that culminated in high feeding damage. The reverse was true for the hairy cultivars. 

Different cassava genotypes expressed different levels of sooty mold damage. This is because different genotypes possess variations in the leaf morphological characteristics. This is re-echoed by [27], showing that cassava leaf area affects sooty mold severity, i.e., a genotype can have high sooty mold severity, yet it harbors a low whitefly population because of its broader leaf surface. However, this contradicts the findings of [13], where they observed no obvious correlation between whitefly population and genotype traits, like leaf width and color.

Sooty mold damage was the greatest in Kamuli and lowest in Lira. This could be associated with the high whitefly populations observed in Kamuli compared to Lira. Previous studies on cantaloupes have associated high whitefly nymph populations with heavy sooty mold establishment in Arizona, United States [34]. Cassava leaf area is another probable factor influencing sooty mold damage levels among different genotypes. According to [27], some cassava genotypes with few whiteflies (*B. tabaci*) supported high sooty mold damage scores probably because of their large leaf area.

Unlike Lira, higher CMD and CBSD incidences were recorded in Wakiso and Kamuli. Disease pressure and whitefly population could explain this observed trend. Studies by [35] found a positive relationship between the whitefly population and disease pressure. The study further attributed the two factors to the unprecedented spread of CBSD in Kamuli and Wakiso. Furthermore, the initial inoculum in the surrounding fields and high vector population influenced CMD incidence in susceptible cassava varieties [36,37]. However, the variation in the reactions of the different genotypes to the viral diseases could be attributed to their inherent genetic makeup. Some genotypes were bred for tolerance or resistance to the two diseases.

## 5. Conclusions

The genotype Mkumba consistently exhibited high levels of resistance to field populations of whitefly (*B. tabaci*) across the three locations. The findings demonstrate the potential of the improved cassava varieties as possible sources of combined disease and whitefly resistance for the sustainable management of the whitefly. This study further recommends the inclusion of Mkumba for participatory variety selection. However, research to ascertain its mode of resistance as well as profiling its biochemical properties should be carried out.

## Figures and Tables

**Figure 1 insects-15-00258-f001:**
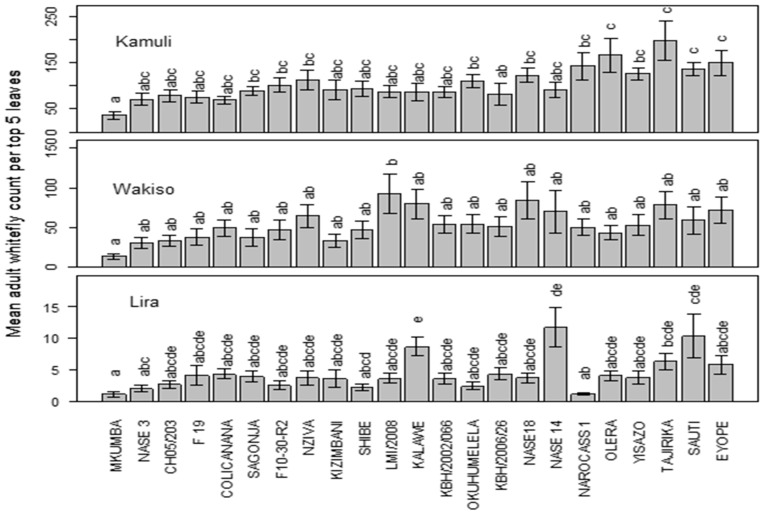
Mean adult whitefly population across the three locations of Kamuli, Wakiso, and Lira. The means followed by the same letter are not significantly different (*p* ≤ 0.05).

**Figure 2 insects-15-00258-f002:**
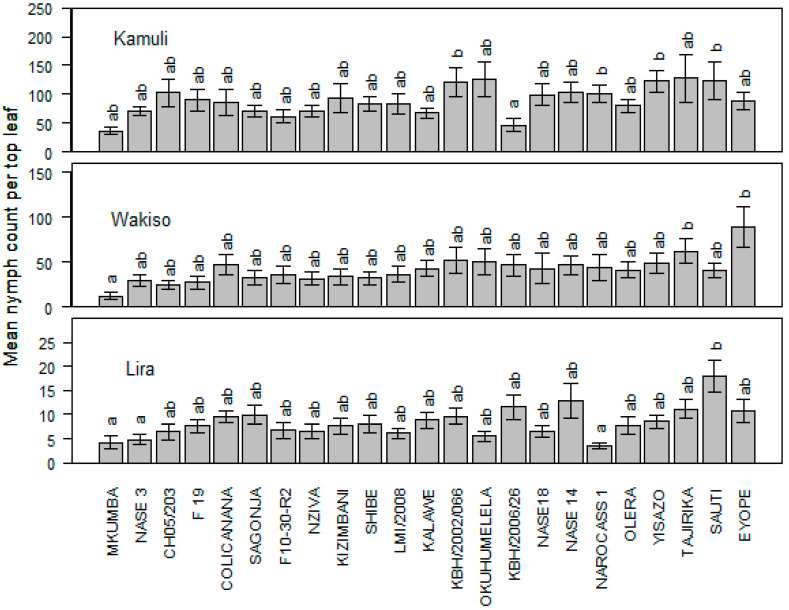
Mean whitefly nymph population across the three locations of Kamuli, Wakiso, and Lira. The means followed by the same letter are not significantly different (*p* ≤ 0.05).

**Table 1 insects-15-00258-t001:** An experiment scheme showing the arrangement of cassava genotypes in the field.

Reps	Cassava Genotypes																						
Rep1	1	2	3	4	5	6	7	8	9	10	11	12	13	14	15	16	17	18	19	20	21	22	23
Rep2	20	23	5	7	1	4	3	17	10	2	15	19	16	18	6	8	9	11	13	21	12	14	22
Rep3	22	3	7	8	9	19	16	4	1	2	17	12	11	5	20	21	23	12	14	15	13	18	10

Rep = replication, 1 = Olera, 2 = Narocass 1, 3 = Nase 3, 4 = Kizimbani, 5 = Mkumba, 6 = Kalawe, 7 = Nziva, 8 = Sagonja, 9 = F19, 10 = Tajirika, 11 = Nase 14, 12 = Nase 18, 13 = Shibe, 14 = CH05/203, 15 = Colicanana, 16 = F10-30 R2, 17 = LM1/2008, 18 = KBH/2002/066, 19 = KBH/2006/026, 20 = Okuhumelela, 21 = Yisazo, 22 = Sauti, and 23 = Eyope.

**Table 2 insects-15-00258-t002:** Mean ± SE whitefly damage and sooty mold severity among cassava genotypes across the two locations of Wakiso and Kamuli. Results are arranged in ascending order with consideration of the district where the highest damage was recorded.

	Whitefly Damage (1–5)				Sooty Mold (1–5)		
Genotype	Wakiso	Kamuli	Lira	Genotype	Kamuli	Wakiso	Lira
KALAWE	1.00 ± 0.00	1.00 ± 0.00	1.00 ± 0.00	KALAWE	1.17 ± 0.17	1.33 ± 0.33	1.00 ± 0.00
KIZIMBANI	1.00 ± 0.00	1.00 ± 0.00	1.00 ± 0.00	KBH/2006/26	1.20 ± 0.20	1.40 ± 0.26	1.00 ± 0.00
SAUTI	1.00 ± 0.00	1.00 ± 0.00	1.00 ± 0.00	OLERA	1.37 ± 0.23	1.67 ± 0.21	1.00 ± 0.00
SHIBE	1.00 ± 0.00	1.33 ± 0.33	1.00 ± 0.00	F10-30-R2	1.50 ± 0.22	1.17 ± 0.17	1.00 ± 0.00
TAJIRIKA	1.00 ± 0.00	0.33 ± 0.33	1.00 ± 0.00	KIZIMBANI	1.50 ± 0.22	1.38 ± 0.25	1.00 ± 0.00
COLICANANA	1.33 ± 0.33	1.67 ± 0.33	1.00 ± 0.00	MKUMBA	1.50 ± 0.22	1.17 ± 0.17	1.00 ± 0.00
F10-30-R2	1.33 ± 0.33	1.00 ± 0.00	1.00 ± 0.00	COLICANANA	1.62 ± 0.28	1.17 ± 0.17	1.00 ± 0.00
KBH/2006/26	1.33 ± 0.33	1.00 ± 0.00	1.00 ± 0.00	EYOPE	1.72 ± 0.23	1.17 ± 0.17	1.00 ± 0.00
LMI/2008	1.33 ± 0.33	1.00 ± 0.00	1.00 ± 0.00	TAJIRIKA	1.75 ± 0.48	1.78 ± 0.27	1.00 ± 0.00
MKUMBA	1.33 ± 0.33	1.00 ± 0.00	1.00 ± 0.00	LMI/2008	1.83 ± 0.31	1.33 ± 0.33	1.00 ± 0.00
OKUHUMELELA	1.33 ± 0.33	1.00 ± 0.00	1.00 ± 0.00	OKUHUMELELA	1.87 ± 0.31	1.67 ± 0.33	1.00 ± 0.00
OLERA	1.33 ± 0.33	1.00 ± 0.00	1.00 ± 0.00	CH05/203	1.90 ± 0.19	1.00 ± 0.00	1.00 ± 0.00
EYOPE	1.66 ± 0.33	1.00 ± 0.00	1.00 ± 0.00	SAGONJA	1.90 ± 0.32	1.33 ± 0.21	1.00 ± 0.00
KBH/2002/066	1.66 ± 0.33	1.33 ± 0.33	1.00 ± 0.00	SAUTI	2.00 ± 0.00	1.17 ± 0.17	1.00 ± 0.00
NASE 14	1.66 ± 0.33	1.67 ± 0.33	1.00 ± 0.00	F 19	2.05 ± 0.26	1.17 ± 0.17	1.00 ± 0.00
NASE18	1.66 ± 0.33	1.80 ± 0.42	1.00 ± 0.00	NZIVA	2.05 ± 0.26	1.33 ± 0.33	1.00 ± 0.00
YISAZO	1.66 ± 0.33	1.00 ± 0.00	1.00 ± 0.00	SHIBE	2.17 ± 0.17	1.40 ± 0.26	1.00 ± 0.00
CH05/203	2.00 ± 0.00	1.00 ± 0.00	1.00 ± 0.00	NASE 14	2.17 ± 0.31	1.00 ± 0.00	1.00 ± 0.00
F 19	2.00 ± 0.00	1.00 ± 0.00	1.00 ± 0.00	NASE18	2.20 ± 0.31	1.45 ± 0.30	1.00 ± 0.00
NAROCASS 1	2.00 ± 0.00	1.00 ± 0.00	1.00 ± 0.00	KBH/2002/066	2.25 ± 0.16	1.79 ± 0.38	1.00 ± 0.00
NASE 3	2.00 ± 0.00	1.00 ± 0.00	1.00 ± 0.00	NAROCASS 1	2.33 ± 0.21	1.50± 0.34	1.00 ± 0.00
NZIVA	2.00 ± 0.00	1.00 ± 0.00	1.00 ± 0.00	NASE 3	2.33 ± 0.33	1.62 ± 0.39	1.00 ± 0.00
SAGONJA	2.00 ± 0.00	1.00 ± 0.00	1.00 ± 0.00	YISAZO	2.43 ± 0.19	1.50 ± 0.22	1.00 ± 0.00

**Table 3 insects-15-00258-t003:** Mean CMD and CBSD incidences among cassava genotypes in Wakiso, Kamuli, and Lira districts. Results are arranged in descending order with consideration of the district where the highest disease incidence was recorded. The means followed by the same letter are not significantly different (*p* ≤ 0.05).

	CMD Incidence (%)				CBSD Incidence (%)		
Genotypes	Wakiso	Kamuli	Lira	Genotypes	Wakiso	Kamuli	Lira
COLICANANA	65.1 a	41.8 a	11.7 a	F 19	43.3 a	5.3 b	0.0 b
NZIVA	58.1 a	39.7 ab	15.1 a	SHIBE	24.1 b	0.0 b	0.0 b
EYOPE	48.9 a	24.5 bc	12.1 a	KALAWE	21.2 bc	0.0 b	10.6 a
NASE 3	26.5 b	12.7 cd	0.7 b	TAJIRIKA	19.4 bcd	5.3 b	0.0 b
OKUHUMELELA	21.7 bc	7.6 d	0.6 b	CH05/203	14.1 bcde	16.1 a	5.1 ab
OLERA	14.0 bcd	1.4 d	0.0 b	LMI/2008	13.3 bcde	0.0 b	0.0 b
KIZIMBANI	13.1 bcd	2.1 d	1.5 b	YISAZO	12.4 bcde	0.0 b	0.0 b
SAUTI	11.6 bcd	0.5 d	0.0 b	SAGONJA	11.8 bcde	0.0 b	0.0 b
F 19	10.9 bcd	3.7 d	0.0 b	SAUTI	6.6 cde	0.0 b	1.0 b
F10-30-R2	10.9 bcd	4.1 d	0.0 b	KBH/2006/26	6.4 cde	0.0 b	0.0 b
CH05/203	10.8 bcd	0.0 d	0.0 b	NASE18	5.5 cde	0.0 b	0.0 b
KBH/2002/066	8.9 bcd	1.4 d	1.9 b	KIZIMBANI	4.5 de	0.0 b	0.0 b
SAGONJA	6.2 cd	2.8 d	1.1 b	COLICANANA	3.7 de	0.0 b	0.6 b
TAJIRIKA	2.3 d	0.0 d	0.0 b	OLERA	3.6 de	1.4 b	0.0 b
LMI/2008	2.1 d	0.0 d	0.0 b	KBH/2002/066	3.6 de	0.0 b	0.0 b
SHIBE	1.4 d	0.3 d	0.0 b	NAROCASS 1	2.3 e	0.0 b	0.0 b
NAROCASS 1	1.4 d	0.0 d	0.0 b	EYOPE	2.1 e	4.2 b	0.0 b
KALAWE	0.5 d	1.4 d	0.0 b	F10-30-R2	1.5 e	0.0 b	0.0 b
YISAZO	0.0 d	0.0 d	0.0 b	NZIVA	1.5 e	1.2 b	0.0 b
KBH/2006/26	0.0 d	0.0 d	0.0 b	NASE 14	0.5 e	0.0 b	0.0 b
MKUMBA	0.0 d	1.6 d	0.8 b	NASE 3	0.4 e	0.0 b	0.0 b
NASE 14	0.0 d	0.0 d	0.0 b	MKUMBA	0.0 e	0.0 b	0.0 b
NASE18	0.0 d	0.0 d	0.0 b	OKUHUMELELA	0.0 e	0.0 b	0.0 b

## Data Availability

Data for this study will be available upon request.

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
