# Peer review of "Resistance to Cassava Whitefly (Bemisia tabaci) among Eastern and Southern African Elite Cassava Genotypes"

_insects, 2024, doi:10.3390/insects15040258_

Round 1

Reviewer 1 Report

Comments and Suggestions for Authors

Dear colleagues,

The authors evaluated the resistance/tolerance of 24 cassava varieties against the white fly vector of two plants viruses in three zones in Uganda. They presented good results; However, I have some concerns regarding the experiments:

-The authors did not do any test (Elisa, RNA extraction…) to verify the presence of virus in plants that did not show any symptoms, or other infected plants.

-I believe if the authors presented environmental data and linked that with population dynamic of Bemisia, it would give further interpretation to the study. Instead, they mentioned some references that studied such effect.

- Another main concern is statistical analysis. The authors handled with count numbers and usually such data is not normally distributed. I suggest that the authors verify again the rawdata and the normality.

More comments in the pdf attached.

Comments on the Quality of English Language

Moderate editing of English language required

Author Response

Response to Reviewer 1 Comments

1. Summary

Much appreciation for the timely review of this manuscript. Detailed responses to the reviewer’s concerns are hereby shared below.

2. Questions for General Evaluation

Reviewer’s Evaluation

Response and Revisions

Does the introduction provide sufficient background and include all relevant references?

Yes/Can be improved/Must be improved/Not applicable

The introduction has been enhanced including rephrasing particular statements as highlighted.

Are all the cited references relevant to the research?

Yes/Can be improved/Must be improved/Not applicable

References have been rephrased especially in text in order to be more relevant.

Is the research design appropriate?

Yes/Can be improved/Must be improved/Not applicable

Research design has been enhanced by providing an experiment design schema.

Are the methods adequately described?

Yes/Can be improved/Must be improved/Not applicable

The materials and methods have been adjusted as highlighted.

Are the results clearly presented?

Yes/Can be improved/Must be improved/Not applicable

Results have been improved by including F values, Degrees of Freedom.

Are the conclusions supported by the results?

Yes/Can be improved/Must be improved/Not applicable

N/A

3. Point-by-point response to Comments and Suggestions for Authors

Comments 1: Abstract needs to be re-written.

Response 1: Grateful for highlighting this. The abstract has been adjusted as per the reviewer’s comments and it’s highlighted in yellow.

Comments 2: Focus on the main objective

Response 2: The abstract is now adjusted and focused towards the main objective

Comments 3: Begin the statement with “In Africa”

Response 3: The introduction is adjusted accordingly as highlighted.

Comments 4: Replace “can be” with “can also be”

Response 4: The introduction is adjusted accordingly as highlighted.

Comments 5: Replace ‘whiteflies’ with “the whitefly”

Response 5: The introduction is adjusted accordingly as highlighted.

Comments 6: Replace ‘the pest’ with “Bemisia tabaci

Response 6: The introduction is adjusted accordingly as highlighted

Comments 7: Rephrase the paragraph in the highlighted introduction section

Response 7: The paragraph in the introduction is adjusted accordingly as highlighted

Comments 8: Begin the paragraph of “the source of planting materials” in the section of materials and methods.

Response 8: The paragraph in the materials and methods is adjusted accordingly as highlighted

Comments 9: Can you provide a schema of your design? I mean how you design the plants genotypes in plots.

Response 9: The schema of the design has been provided in the section of materials and methods as highlighted.

Comments 10: How you evaluated if the plant was infected by the two viruses?

Response 10: The same scoring scales for individual viruses (Cassava Mosaic and Cassava Brown Streak) were applied as stated in the manuscript, especially when the plant had dual viral infection.

Comments 11: There is an issue here, where count data does not follow the normality. You need to verify the raw data.

Response 11: We used the model (quasipoison) that accounts for data that is not normally distributed.

Comments 12: P values are not enough. You have to mention the F values and degrees of freedom. Idem for other results.

Response 12: F values and degrees of freedom have been included as per highlighted text.

Comments 13: Mean +- SE. Idem for table 2

Response 13: This change has been effected as per highlighted text.

Comments 14: What are these parameters?

Response 14: These weather parameters are listed in detail as per highlighted text

Comments 15: Do you mean your study or the previous citation? Rephrase

Response 15: The statement is rephrased accordingly as per highlighted text in the manuscript.

Comments 16: Ok, but have you surveyed and observed natural enemies on the plants

Response 16: We simply observed the natural enemies in our trials but didn’t carry out specialized surveys. However, there are publications confirming the action of predators on whitefly population but these are not specific to the areas the study was carried out.

Comments 17: But in line 240: you mentioned that Kamuli had the high whitefly numbers.

This sentence with the following one may need to be rephrased

Response 17: The statement has been rephrased as highlighted in the manuscript.

4. Response to Comments on the Quality of English Language

Point 1: Moderate editing of English language required

Response 1: This was done and it included rephrasing some statements to make the paper clearer for the audience to understand.

5. Additional clarifications

From previous studies, the viral species present in the different locations of the study are known and been documented. Therefore, there was no need to carry out ELISA tests.

The weather data for specific areas where the study had been conducted was not collected. However, we utilized the general agroecological weather conditions as described under the experimental sites.

Reviewer 2 Report

Comments and Suggestions for Authors

Dear Authors,

The research has some interesting aspects, it is written in a clear and easily understandable manner and one can see that it took a lot of work to complete it. However, there are some obscure points, some rather serious omissions. For example, it is not specified for how long the observations were conducted. This is important, because the production of variety lines is continuous and some of those that were on the market eight years ago may have been withdrawn today. Furthermore, resistance is a mechanism that can be overcome by insects in a short time. Consequently, some varieties that were valid 8 years ago, may no longer be valid today. The statistical analysis is also insufficient, as no analysis was performed to verify the comparison and whether the data are significant or not. The publication has merits, but also many flaws. For me it should be strongly revised.

Author Response

Response to Reviewer 2 Comments

1. Summary

Much appreciation for the timely review of this manuscript. Detailed responses to the reviewer’s concerns are hereby shared below.

2. Questions for General Evaluation

Reviewer’s Evaluation

Response and Revisions

Does the introduction provide sufficient background and include all relevant references?

Yes/Can be improved/Must be improved/Not applicable

The abstract and introduction was adjusted as highlighted

Are all the cited references relevant to the research?

Yes/Can be improved/Must be improved/Not applicable

Most of the references are relevant to the research.

Is the research design appropriate?

Yes/Can be improved/Must be improved/Not applicable

The design was appropriate and was slightly adjusted as highlighted.

Are the methods adequately described?

Yes/Can be improved/Must be improved/Not applicable

They are adequate but with slight adjustment as highlighted.

Are the results clearly presented?

Yes/Can be improved/Must be improved/Not applicable

More robust statistical analyses were carried out and F values, DF, mean separation values given in the text.

Are the conclusions supported by the results?

Yes/Can be improved/Must be improved/Not applicable

We believe that the conclusions are now well supported by the results since there has improvement in statistical analysis. Also, some statements were rephrased in the discussion section  for clarity.

3. Point-by-point response to Comments and Suggestions for Authors

Comments 1: It is advisable to mention what these acronyms correspond to, someone might not know and would have to go looking for it.

Response 1: Grateful for highlighting this. The abstract has been adjusted as per the reviewer’s comments and it’s highlighted in yellow.

Comments 2: As reported by (7)

Response 2: This has been effected in the introduction section as per highlighted text.

Comments 3: Sorry, the experiment started in 2016 and when did it end? In the same year, after 1, 2, 4 years or is it still ongoing? It is not clear. Also, because if the study was conducted in 2016 (8 years ago!) what is the point? New varieties are constantly being produced and some of the old ones may be out of market.

Response 3: The experiment was evaluated for 1 year. The findings provided in this paper actually guided the breeders to include this best performing variety, Mkumba in the crossing block so as to tap the genes of whitefly resistance into the breeding scheme. Right now, these clones are in advanced stages of selection.

Comments 4: I imagine that the count was carried out in the very early hours of the morning, considering that when temperatures rise, Bemisia adults fly away very quickly. It would have been interesting to also perform an egg count, because some varieties are tolerant due to a particular tomentose structure, or a volatile substance that drives away insects.

Response 4: In addition to adults, we assessed the nymphal stages that are not mobile at all, thus stable. The data presented has both adults and nymphs (immature stages of whitefly).

Comments 5: How do you explain the fact that Mkumba was the variety with the lowest number of adults and neanids, but with damage and smut more or less at the level of the other varieties? Why there is any statistical analysis to compare the results?

Response 5: Firstly, expression of feeding damage and sooty mold symptoms is not only influenced by the whitefly population but also leaf morphology. Varieties with thick cassava leaves might express less damage symptoms even though they harbor high populations. As for Statistical analysis, looking at the severity scoring scale of 1-5, most of the symptoms observed were within the lower limit (Score 1- Score 2) which is a low score to give significant differences.

Comments 6: Why there is no statistical analysis to compare the significance of the differences?

Response 6: More statistical analysis has been performed to include the genotype respective means separated by differences in letters denoting significance as reflected in the text.

Comments 7: I think there is some punctuation problem. Perhaps it is better to rephrase the period.

Response 7: The statement has been rephrased as highlighted in the text.

Comments 8: There is no fine if sometimes the name(s) of any author(s) of a scientific publication is mentioned!

Response 8: The statement has been adjusted as highlighted.

Round 2

Reviewer 1 Report

Comments and Suggestions for Authors

The authors improved their MS. I still have few comments:

1-The authors stated that the viral species were present in the different locations and documented. Therefore, there was no need to carry out ELISA test. I dont agree with this. Better to say that you did not test the presence of virus in plants.

2- Since you did not have the weather data, dont talk much about the weather in discussion, to avoid the overinterpretation of the present data.

3- Regarding the experimental protocol, I was wondering if 3 replicates are enough 

4-for degree of freedom, start with small number then the total number.

L194: delete: " the data were subjected to ANOVA "

L343-350: needs a ref.

L339: mention the name of ref (23)

Comments on the Quality of English Language

Check the E language

Reviewer 2 Report

Comments and Suggestions for Authors

Dear Authors,

Thank you for collecting the suggestions, improving the text and adding further statistical analysis. The research, as far as I'm concerned, can be published. Congratulations.

Best regards

Author Response

Thank you so much for your valuable time to review this work